# Understanding Language in the World by Predicting the Future

## Abstract

To interact with humans and act in the world, agents need to understand the range of language that people use and relate it to the visual world. While current agents can learn to execute simple language instructions, we aim to build agents that leverage diverse language—language like "this button turns on the TV" or "I put the bowls away"—that conveys general knowledge, describes the state of the world, provides interactive feedback, and more. Our key idea is that *agents should interpret such diverse language as a signal that helps them predict the future*: what they will observe, how the world will behave, and which situations will bring high reward. This perspective unifies language understanding with future prediction as a powerful self-supervised learning objective. We instantiate this in Dynalang, an agent that learns a multimodal world model to predict future text and image representations, and learns to act from imagined model rollouts. Unlike current agents that use language to predict actions only, Dynalang acquires a rich language understanding by learning to predict future language, video, and rewards. In addition to learning from online interaction in an environment, we show that Dynalang can be pretrained on text-only datasets, enabling learning from more general, offline datasets. From using language hints in grid worlds to navigating photorealistic home scans, Dynalang can leverage diverse types of language, e.g. environment descriptions, game rules, and instructions.

## 1 Introduction

A long-standing goal of artificial intelligence is to develop agents that can use language to interact naturally with people in the physical world (Winograd, 1972). Current embodied agents can follow simple, low-level instructions like "get the blue block" (Lynch et al., 2022) or "go past the elevator and turn right" (Anderson et al., 2018). However, to communicate freely interactive agents should understand the full range of ways people use language beyond the "here and now" (Hockett & Hockett, 1960): transmitting knowledge such as "the top left button turns off the TV," providing situational information such as "we're out of milk," and coordinating by saying "I already vacuumed the living room." Much of what we read in text or hear from others communicates *knowledge about the world*, either about how the world works or about the current state of the world.

How could we enable agents to use such diverse types of language? One way to train language-conditioned agents to solve tasks is reinforcement learning (RL). However, current language-conditioned RL methods primarily learn to generate actions from task-specific language instructions, e.g. taking a goal description like "pick up the blue block" as an input and outputting a sequence of motor controls that achieve high reward. When we consider the diversity of functions that natural language serves in the real world, directly mapping language to optimal actions presents a challenging learning problem. Consider the example "I put the bowls away": if the task at hand is cleaning up, the agent should respond by moving on to the next cleaning step, whereas if it is serving dinner, the agent should retrieve the bowls. Thus, "I put the bowls away" is not an instruction of what task to do, rather it is a description of the world, to be interpreted in the context of the current task. When language does not talk about the task, it is only weakly correlated with optimal actions the agent should take. Mapping language to actions, particularly using task reward alone, is therefore a weak learning signal for learning to use diverse language inputs to accomplish tasks.

*Instead, we propose that a unifying way for agents to use language is as signal that helps them predict the future.* This is our core contribution. The utterance "I put the bowls away" helps agents make better predictions about future observations (i.e., that if it opens the cabinet, it will observe the bowls

**Context**

Video and text inputs

**Dynalang Model Rollouts**

Video prediction

**Reward prediction**

r=0   r=0   r=0   r=0   r=1

**Text prediction**

kitchen

Figure 1: Dynalang learns to use language to make predictions about future (text + image) observations and rewards. Here, we show real model predictions in the HomeGrid environment. From the past text "the bottle is in the living room", the agent predicts at timesteps 61-65 that it will see the bottle in the final corner of the living room. From the text "get the bottle" describing the task, the agent predicts that it will be rewarded for picking up the bottle. The agent can also predict future text observations: given the prefix "the plates are in the" and the plates it observed on the counter at timestep 30, the model predicts the most likely next token is "kitchen."

there). Much of the language we encounter can be grounded in visual experience in this way. Prior knowledge such as "wrenches can be used to tighten nuts" helps agents predict environment dynamics. We can also formulate instruction following predictively—instructions help agents predict how they will be rewarded. Similar to how next-token prediction allows language models to form internal representations of world knowledge (Petroni et al., 2019), we hypothesize that predicting future representations provides a rich learning signal for agents to understand how language is grounded in the world. In turn, agents can leverage this understanding by planning or learning to act against their language-enabled predictive model.

We present Dynalang, an agent that learns a world model of language and images from online experience and uses the model to learn how to act. Dynalang operationalizes our proposal—using language to predict the future—by learning a world model that compresses both visual and textual inputs to a latent space. The world model is trained to predict future latent representations with experience collected from acting in the environment, effectively learning an simulator of how the environment behaves. We instantiate this by building on the DreamerV3 (Hafner et al., 2023) architecture, but show that naive ways of aligning language, vision, and action from prior works are not effective. Dynalang learns a sequence model over image frames and text tokens at each timestep, both enabling an interface where the agent can read, listen, and speak while simultaneously receiving new visual inputs and taking motor actions and empirically outperforming other ways of learning world models conditioned on language. Because world modeling is separate from policy training, Dynalang can also be pretrained on single modalities (text-only or video-only data) without actions or task reward. We demonstrate the efficacy of text-only pretraining, allowing Dynalang to leverage general-domain text data. Additionally, *language generation* can also be unified in our framework: the agent's perception can inform its language model (i.e., future token predictions), which can enable it to speak about the environment by outputting language as actions.

We evaluate Dynalang on a range of domains with different types of language. We construct a home cleanup environment with diverse language types, HomeGrid, where Dynalang learns to use language hints about future observations, environment dynamics, and corrections. On the Messenger benchmark (Hanjie et al., 2021), Dynalang can read game manuals to fit the most challenging stage of the game, outperforming task-specific architectures. In vision-language navigation (Krantz et al., 2020), Dynalang can follow instructions in visually and linguistically complex domains.

Our contributions are as follows:

- We propose that RL agents can learn to ground diverse types of language to visual experience by predicting the future in a world model.
- We implement Dynalang to instantiate this idea. We show that Dynalang learns to understand diverse kinds of language to solve a broad range of tasks, often outperforming other language-conditioned model-based agents and state-of-the-art RL algorithms.
- We show that this formulation enables additional capabilities: *language generation* can be unified in the same model, as well as text-only pretraining without actions or task rewards.

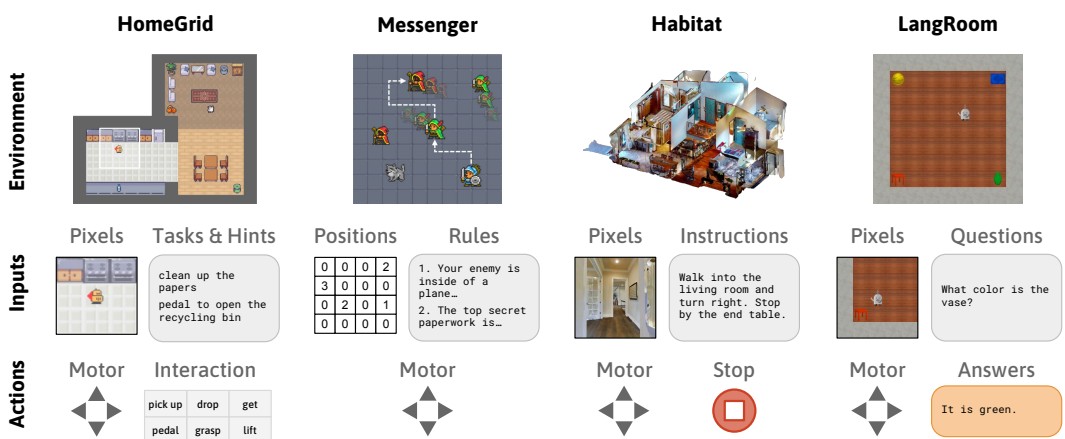

Figure 2: We consider a range of environments that feature visual inputs and diverse types of language. We introduce HomeGrid, a challenging visual gridworld with instructions and diverse hints. Messenger is a benchmark with symbolic inputs and hundreds of human-written game manuals that require multi-hop reasoning. Habitat simulates photorealistic 3D homes for vision-language navigation in hundreds of scenes. LangRoom is a simple visual grid world with partial observability, where the agent needs to produce both motor actions and language.

## 2 RELATED WORK

Much work has focused on teaching reinforcement learning agents to utilize language to solve tasks by directly conditioning policies on language (Lynch & Sermanet, 2021; Shridhar et al., 2022; Abramson et al., 2020). More similar to our work, recent work proposes text-conditioning a video model trained on expert demonstrations and using the model for planning (Du et al., 2023a; Yang et al., 2023). However, language in these settings has thus far been limited to short instructions, and only a few works investigate how RL agents can learn from other kinds of language like descriptions of how the world works, in simple settings (Zhong et al., 2020; Hanjie et al., 2021).

In reality, human language is far richer than imperative commands. Other work looks to diversify the utterances that agents can understand with supervised learning on human-annotated or expert datasets in embodied environments, e.g. to understand more complex instructions (Ku et al., 2020; Shridhar et al., 2020a) or environment descriptions (Zhong et al., 2022), ask for assistance in simulated navigation or household tasks (Thomason et al., 2019; Padmakumar et al., 2022; Abramson et al., 2020), answer questions (Das et al., 2018), or collaborate dynamically on a shared goal (Bara et al., 2021). Alternatively, recent work has proposed augmenting embodied agents with pretrained large language models (LLMs) to endow them with more general language capabilities, e.g. by breaking down complex language context into simple instructions for low-level policies (Huang et al., 2022a; Li et al., 2022; Ahn et al., 2022), integrating knowledge from text (Wu et al., 2023), or directly serving as the policy (Driess et al., 2023; Wang et al., 2023; Carta et al., 2023). These works approach the realism of natural language in the diversity of utterances and *roles* of language in the world they consider. However, supervised approaches suffer from reliance on expensive human data (often with aligned language annotations and expert demonstrations), and both these approaches have limited ability to improve their behaviors and language understanding online.

Our goal is to propose a framework for unifying these paradigms with an agent that can learn from both autonomous exploration with RL and diverse types of language in the world. In contrast to previous RL agents, Dynalang takes a step towards more diverse language understanding by making predictions in a world model, providing a learning signal that is both rich (unlike rewards) and self-supervised (unlike demonstrations). In the context of supervised approaches, many of which also architecturally learn language-conditioned *policies*, our work suggests that learning a world model of language and vision may be a more scalable approach that enables autonomous improvement with the familiar benefits of supervision such as rich learning signal and pretraining. However, for the most part, we focus on a specific way of scaling up language complexity—controlled experiments using different *types* of scripted language—leaving future work to fully bridge the gap to natural language in real-world human interaction. We refer to Appendix C for a more detailed discussion of related work.

## 3 DYNALANG

Dynalang utilizes diverse types of language in visual environments by encoding multiple modalities into learned representations and then predicting the sequence of future representations given actions. For our algorithm, we build on the model-based algorithm DreamerV3 (Hafner et al., 2023) and extend it to process and optionally produce language. The world model is trained from a replay buffer of past experience while the agent is interacting with the environment. It can additionally be pretrained from text-only data. To select actions, we train an actor-critic model from sequences of representations imagined by the world model. The algorithm is summarized in Algorithm 1.

**Problem setting** To perform interactive tasks, an agent chooses actions $a_t$ to interact with an environment that responds with rewards $r_t$, a flag for whether the episode continues $c_t$, and observations $o_t$. In this paper, we consider multimodal environments where $o_t = (x_t, l_t)$ consist of an image $x_t$ and a language token $l_t$ at each time step. The agent's goal is to choose actions that maximize the expected discounted sum of rewards $\mathrm{E}\left[\sum_{t=1}^{T} \gamma^t r_t\right]$, where $\gamma < 1$ is a discount factor, $T$ is the episode length, and $c_T = 0$ signals the episode end. In most of our experiments, the actions $a_t$ are integers in a categorical action space. However, we also consider factorized action spaces where the agent outputs both a discrete movement command and a language token at each time step.

**Multimodal alignment** We consider a diverse range of environments, summarized in Figure 2, where agents receive a continuous stream of video and text observations. While

---

**Algorithm 1:** Dynalang

**while** *acting* **do**
  Step environment $r_t, c_t, x_t, l_t \leftarrow \mathrm{env}(a_{t-1})$.
  Encode observations $z_t \sim \mathrm{enc}(x_t, l_t, h_t)$.
  Execute action $a_t \sim \pi(a_t \mid h_t, z_t)$.
  Add transition $(r_t, c_t, x_t, l_t, a_t)$ to replay buffer.

**while** *training* **do**
  Draw batch $\{(r_t, c_t, x_t, l_t, a_t)\}$ from replay buffer.
  Use world model to compute multimodal representations $z_t$, future predictions $\hat{z}_{t+1}$, and decode $\hat{x}_t, \hat{l}_t, \hat{r}_t, \hat{c}_t$.
  Update world model to minimize $\mathcal{L}_{\mathrm{pred}} + \mathcal{L}_{\mathrm{repr}}$.
  Imagine rollouts from all $z_t$ using $\pi$.
  Update actor to minimize $\mathcal{L}_\pi$.
  Update critic to minimize $\mathcal{L}_V$.

**while** *text pretraining* **do**
  Sample text batch $\{l_t\}$ from dataset.
  Create zero images $x_t$ and actions $a_t$.
  Use world model to compute representations $z_t$, future predictions $\hat{z}_{t+1}$, and decode $\hat{l}_t$.
  Update world model to minimize $\mathcal{L}_{\mathrm{pred}} + \mathcal{L}_l$.

---

previous settings specify that language such as instructions arrive at the beginning of an episode, we are interested in enabling agents to act in more flexible settings where they face a continuous stream of video and text, as in the real world. For humans, reading, listening, and speaking extends over time, during which we receive new visual inputs and can perform motor actions. Analogously, we provide our agent with one video frame and one language token at each time step and the agent produces one motor action, and in applicable environments one language token, per time step. We show that this design outperforms other ways of incorporating language in Section 4.1. An additional benefit of providing one token per time step is that the algorithm does not have to decide on an arbitrary way to segment language temporally. We show in Appendix F.2 that token-level representations substantially outperform sentence-level representations.

### 3.1 WORLD MODEL LEARNING

The world model learns representations of all sensory modalities that the agent receives and then predicts the sequence of these latent representations given actions. Predicting future representations not only provides a rich learning signal to ground language in visual experience but also allows planning and policy optimization from imagined sequences. The world model is shown in Figure 3a. At each time step, it receives an image $x_t$, a language token $l_t$, and an action $a_t$. The image and language observations are compressed into a discrete representation $z_t$ and fed together with the action into the sequence model to predict the next representation $\hat{z}_{t+1}$. The multimodal world model consists of the following components, where $h_t$ is a recurrent state:

$$\text{Sequence model:} \qquad \hat{z}_t, \, h_t = \mathrm{seq}(z_{t-1}, \, h_{t-1}, \, a_{t-1})$$

$$\text{Multimodal encoder:} \qquad z_t \sim \mathrm{enc}(x_t, \, l_t, \, h_t)$$

$$\text{Multimodal decoder:} \qquad \hat{x}_t, \, \hat{l}_t, \, \hat{r}_t, \, \hat{c}_t = \mathrm{dec}(z_t, \, h_t)$$

We implement the world model as a Recurrent State Space Model (RSSM Hafner et al., 2018), where the sequence model is implemented as GRU (Cho et al., 2014) with recurrent state $h_t$. Using

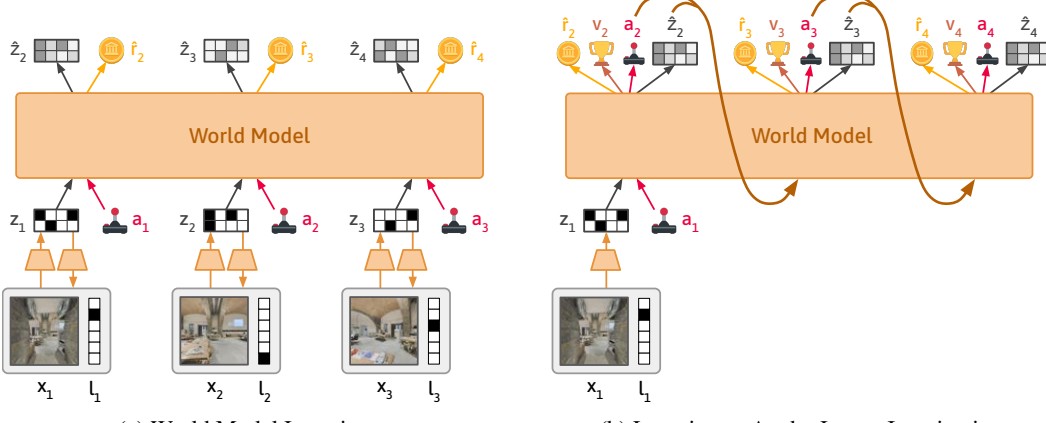

(a) World Model Learning       (b) Learning to Act by Latent Imagination

Figure 3: During world model learning, the model compresses observations of image frames and text to a latent representation. The model is trained to predict the next representation and reconstruct observations from the representation. During policy learning, imagined rollouts are sampled from the world model and the policy is trained to maximize imagined rewards.

a recurrent model has the benefit that the policy does not have to integrate information over time anymore, but other sequence models such as Transformers can also be used (Chen et al., 2022a; Robine et al., 2023). At each timestep, the encoder conditions on the observations and model state $h_t$, effectively learning to compress observations to codes $z_t$ relative to the history. The sequence model then conditions on the encoded observations $z_t$ to integrate new observations into the next model state. The decoder is trained to reconstruct observations and other information, thus shaping the model representations. The world model is trained jointly to minimize a representation learning loss $\mathcal{L}_{\text{repr}}$ and a future prediction loss $\mathcal{L}_{\text{pred}}$, which we describe below.

**Multimodal representations** The world model learns to compress inputs images $x_t$ and language tokens $l_t$ into stochastic latent representations $z_t$ through a variational autoencoding objective (Kingma & Welling, 2013; Rezende et al., 2014). The representations are shaped by reconstructing the input observations, providing a rich learning signal for grounding. We also predict the reward, $\hat{r}_t$, and whether the episode continues, $\hat{c}_t$, so that the policy can be learned directly on top of the latent representations, as discussed in the next section. Finally, the representations are regularized towards the predicted distribution over $\hat{z}_t$ as a prior, essentially regularizing the representations to be predictable. We denote the categorical cross entropy loss as catxent, the binary cross entropy loss as binxent, the stop gradient operator as sg, and $\beta_{\text{reg}} = 0.1$ is a hyperparameter.

The representation learning loss $\mathcal{L}_{\text{repr}}$ is thus the sum of terms:

$$\begin{array}{ll} \text{Image loss:} & \mathcal{L}_x = \|\hat{x}_t - x_t\|_2^2 \\ \text{Language loss:} & \mathcal{L}_l = \text{catxent}(\hat{l}_t, l_t) \\ \text{Reward loss:} & \mathcal{L}_r = \text{catxent}(\hat{r}_t, \text{twohot}(r_t)) \\ \text{Continue loss:} & \mathcal{L}_c = \text{binxent}(\hat{c}_t, c_t) \\ \text{Regularizer:} & \mathcal{L}_{\text{reg}} = \beta_{\text{reg}} \max(1, \text{KL}\big[z_t \,\|\, \text{sg}(\hat{z}_t)\big]) \end{array}$$

We choose a strided CNN image encoder, a strided CNN as image decoder, and MLPs for all other model components. We evaluate our method both with one-hot token observations (i.e., learning the embeddings from scratch) and pretrained embeddings from T5 (Raffel et al., 2020). One-hot representations are reconstructed with the cross entropy loss above and pretrained embeddings are reconstructed with a squared error. For more details on world model learning, refer to Appendix A.

**Future prediction** The world model learns to predict the sequence of multimodal representations, which enables it to plan and ground language. The sequence model produces $\hat{z}_t$ from the current model state $(z_{t-1}, h_{t-1})$ and the current action $a_{t-1}$, whih is trained to match the actual representation at the next timestep $z_t$. Concretely, the future prediction objective is:

$$\text{Prediction loss:} \quad \mathcal{L}_{\text{pred}} = \beta_{\text{pred}} \max(1, \text{KL}\big[\text{sg}(z_t) \,\|\, \hat{z}_t\big])$$

where the gradient around the target distribution for $z_t$ is stopped since it is also a learned representation and $\beta_{\text{pred}} = 0.5$ is a hyperparameter. Intuitively, the codes $z_t$ contain information from current observation, but also additional information that may be required to predict the reward and episode

continuation. By training the world model to make predictions $\hat{z}_t$ of its future representations, it effectively learns to predict future images, language, and rewards from its inputs, encouraging the agent to extract information from language and learn the correlations between its multiple modalities. For example, when the language input describes that "the book is in the living room" and the agent later on visually observes the book, the agent will learn this multimodal association even if the reward signal does not directly relate the two. This objective provides a rich learning signal for grounding. The world model is trained to optimize the overall loss $\mathcal{L}_{\text{repr}} + \mathcal{L}_{\text{pred}}$ with respect to all its parameters.

**Single-Modality Pretraining**  One potential benefit of separating world modeling from policy learning is that the world model can be trained offline, benefitting from large-scale text-only and video-only datasets without actions. To pretrain the world model with text-only data as in Section 4.6, we zero out the image and action inputs and set the image, reward, and continuation decoder loss coefficients to 0 so the pretraining focuses on learning to represent text and text dynamics (i.e. language modeling). Dynalang can then be finetuned on experience with all modalities (language, images, and actions) by initializing the actor and critic from scratch, while continuing to train the world model. Note that unlike the typical language modeling objective, the model is not explicitly trained to predict the next token from the prefix, except through the next-*representation* prediction.

### 3.2  POLICY LEARNING

To select actions, we train an actor critic algorithm (Williams, 1992) purely from imagined sequences of multimodal representations (Sutton, 1991). The purpose of the critic is to estimate the discounted sum of future rewards for each state to guide the actor learning. Both networks are MLPs:

$$\text{Actor network:} \quad \pi(a_t|h_t, z_t) \qquad \text{Critic network:} \quad V(h_t, z_t)$$

We do not modify the policy learning algorithm of DreamerV3 and refer to Appendix B for details. During environment interaction, we sample actions from the actor without planning.

## 4  EXPERIMENTS

Our experiments aim at investigating the following hypotheses:

**H1)** Aligning image and language observations as single (image, token) pairs per timestep outperforms other methods for incorporating language into DreamerV3 (Section 4.1).

**H2)** Dynalang enables agents to leverage language beyond instructions to improve task performance. To test this, we investigate whether adding different kinds of language hints in HomeGrid improves task performance (Section 4.2), and whether Dynalang can learn from game manuals in Messenger (Section 4.3).

**H3)** It is more useful to ground diverse language with the future prediction objective than to predict actions directly. To test this, we compare our method against model-free RL baselines.

**H4)** Interpreting instructions as future reward prediction is no worse than learning to predict actions directly from instructions, as is typically done. To test this, we compare performance to baselines with task-only language in HomeGrid and on vision-language navigation (Section 4.4).

**H5)** Dynalang additionally enables the agent to generate language (Section 4.5) and enables text-only pretraining, allowing our agents to learn from general, single-modality datasets (Section 4.6).

**Language encodings**  We tokenize all text with the T5 tokenizer (Raffel et al., 2020), with a vocabulary size of 32,100. In HomeGrid we use one-hot token encodings. In Messenger and VLN-CE, where agents must generalize to synonyms and linguistic variations, we embed each sentence with T5-small (60M parameters) and use the last hidden layer representation for each token.

**Baselines**  We compare against two off-policy model-free RL baselines: IMPALA (Espeholt et al., 2018) and R2D2 (Kapturowski et al., 2019). The architecture for both algorithms consists of an LSTM that takes in input embeddings from a CNN image encoder and an MLP language encoder. We use the implementations from the SeedRL repository (Espeholt et al., 2019). We pass the same language observations to the baselines as to our method (token embeddings or one-hot encodings). We also try providing the baselines with sentence embeddings from a pretrained `all-distilroberta-v1` model from the Sentence Transformers library (Reimers & Gurevych, 2019) and did not find a consistent improvement across our tasks. Both baseline models are ∼10M parameters, and we did not find that these models benefit from scaling parameter count.

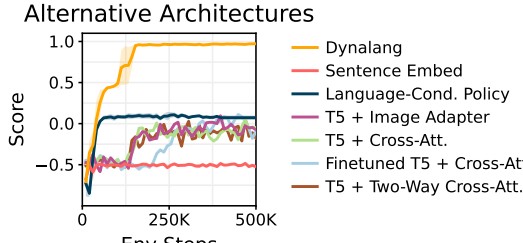

Figure 4: Comparison of ways to equip the world model with language inputs on Messenger S1. We compare ways of conditioning DreamerV3 on language and find that the Dynalang architecture substantially outperforms other approaches, even those based on pretrained T5, despite training from scratch.

### 4.1 ALIGNING LANGUAGE, VISION, AND ACTION IN A WORLD MODEL

What is an effective way to model language in a setting where agents are acting online in an environment with a stream of visual observations? Previous work on language-conditioned RL typically encodes a single sentence representing the task. However, in more complex language settings, agents may receive inputs such as long manuals and interactive feedback over time. We are proposing a general method to model diverse types of language by conditioning on a stream of text tokens in a world model, interleaved with visual observations. We now evaluate this design choice by comparing Dynalang to alternative language-conditioned DreamerV3 agents, isolating the effect of our design choices for language inputs from the base DreamerV3 architecture.

We use Messenger's Stage 1 as a testbed, where the agent must learn to read a text manual in order to achieve high reward (see Section 4.3 for details). We implement several common language conditioning methods from previous work on multimodal models and language-conditioned RL. Figure 4 compares Dynalang to five other ways of integrating language inputs to the DreamerV3 base agent:

1. **Language-Conditioned Policy** Embed tokens with a GRU and condition the policy on the final hidden state. This baseline is similar to how model-free approaches must implement language conditioning (Shridhar et al., 2022; Lynch et al., 2023), but the agent is still learns to model image observations with a world model. This ablates the effect of learning joint representations of language and images together in the world model.

2. **Sentence Embed** Input text into the world model one sentence at a time, using Sentence-BERT (Reimers & Gurevych, 2019) to embed each sentence of the manual. This ablates the effect of inputting one token per timestep.

3. **T5 with Image Adapter** Train the image encoder to map image features to the embedding space of a pretrained T5 language model, following multimodal architectures such as LLaVA (Liu et al., 2023). This allows the agent to use the powerful pretrained representations of the language model for multimodal representations. We map the image at each timestep to 10 language tokens with an MLP, attend over the image tokens and the tokens of the language input with a frozen T5 encoder, and condition the world model on the last hidden state of T5 as the multimodal representation.

4. **T5 with Cross-Attention** Embed the entire manual with T5 and then fuse with the image observation by attending to the sequence of token embeddings with the image embedding output by the image encoder, similar to the original Messenger baseline (Hanjie et al., 2021).

5. **Finetuned T5 with Cross-Attention** (3.), but finetune the T5 encoder.

6. **T5 with Two-Way Cross-Attention** (3.), but additionally map images to a fixed number of latents and attend to it with the pooled text embedding, following multimodal models such as Lu et al. (2016).

We find that Dynalang outperforms these alternatives even with token embeddings initialized from scratch while also being simple and efficient to train, supporting **H1**. While most other language-conditioned agents embed one sentence at a time, our comparison shows that these approaches underperform our approach in Dynalang of encoding one token per timestep. This architecture is also analogous to language model architectures, enabling us to pretrain the same agent architecture on text-only data in the same way in Section 4.6.

### 4.2 HOMEGRID: LANGUAGE HINTS

As most standard RL benchmarks do not provide language beyond instructions, we introduce a new environment, HomeGrid, that evaluates how well agents can ground diverse types of language to

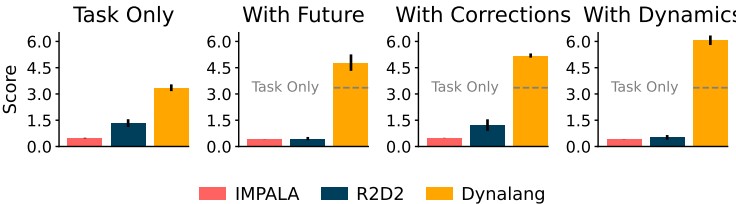

Figure 5: HomeGrid performance after 50M steps (2 seeds). Dynalang learns to use all types of language hints to score higher than when just provided with the task information, outperforming language-conditioned IMPALA and R2D2, where we see performance decrease with language hints.

solve tasks. HomeGrid is a multitask gridworld where agents receive language task specifications and *language hints*, as depicted in Figure 6. Hints provide prior knowledge about world dynamics, information about world state, or corrections that assist the agent. The agent can otherwise acquire the same information through its own interaction with the environment, as in standard RL. Agents can achieve higher performance if they learn to ground language.

There are five task types involving objects and trash bins (find, get, clean up, rearrange, open) for a total of 38 tasks. Agents get pixel observations with a partially observed view of the environment and can move and interact with objects and trash bins. Object locations, bin locations, and bin dynamics (*i.e.,* which action correctly opens the bin) are randomized on each episode. Objects are also randomly moved throughout the episode. Agents receive task specifications in language. When a task is completed, the agent gets a reward of 1 and a new task is sampled. To achieve a high score, agents must complete as many tasks as possible before the episode terminates in 100 steps. We also provide hints at random points throughout the episode that are provided token-by-token while the agent continues to act. We script the following language hints:

- **Future observations** Descriptions of where objects are in the world or where they have been moved. Without language, the agent must explore the environment to find objects.
- **Dynamics** Descriptions of the correct action to open each trash bin. Without language, the agent can try all the different actions, although taking the wrong action can disable the trash can for a variable number of timesteps or potentially the rest of the episode (irreversible dynamics).
- **Corrections** Tell the agent "no, turn around" when its distance to the current goal object is increasing. Without language, the agent must explore on its own.

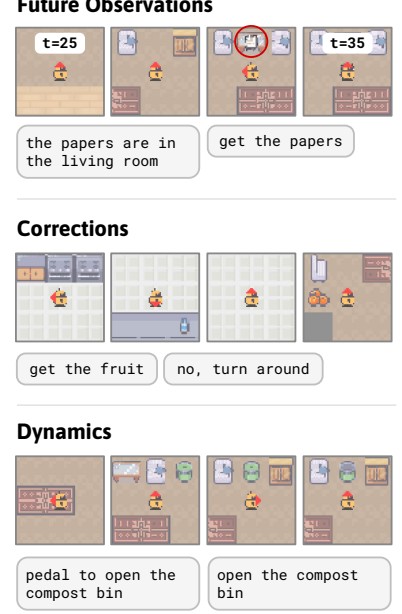

Figure 6: HomeGrid provides language hints and task specifications. We show real trajectories from a trained agent.

Figure 5 shows that Dynalang benefits from all kinds of language, achieving higher scores with hints relative to just using instructions. Notably, agents *never receive direct supervision* about what the hints mean in HomeGrid, and hints are often far removed from the objects or observations they refer to. Dynalang learns to ground language to the environment purely via the future prediction objective. Language-conditioned IMPALA struggles to learn the task at all, while R2D2 learns to use the types of language that are correlated with reward (tasks and corrections). Interestingly, we find that while R2D2's performance drops as it gets overwhelmed with more diverse language, while Dynalang improves across the board, supporting **H2** and **H3**. We hypothesize that additional language input makes it more difficult for the model-free methods to learn to process observations to solve the task.

### 4.3 MESSENGER: GAME MANUALS

Next, we evaluate Dynalang on the Messenger game environment (Hanjie et al., 2021), which tests whether agents can read text manuals describing game dynamics to achieve high scores. In

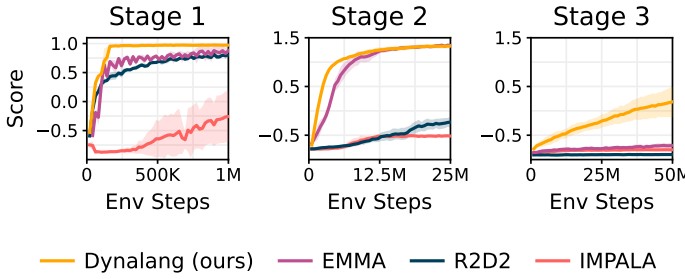

Figure 7: Messenger training performance (2 seeds). Dynalang outperforms language-conditioned IMPALA and R2D2, as well as the task-specific EMMA architecture, fitting the most complex stage of the game where other methods fail to achieve non-trivial performance.

Messenger, the agent must retrieve a message from one of the entities in the environment and deliver it to another entity, while avoiding enemies. In each episode, the agent receives a manual describing the randomized entity roles and movement dynamics. The challenge is grounding the text references to the environment, which requires multi-hop reasoning over both visual and text inputs (e.g. combining the manual information that the goal entity is a "fleeing wizard" with observations of entity identities and movement dynamics). Messenger has three stages of increasing length and difficulty.

In addition to the baselines above, we compare the performance of Dynalang to EMMA, the original baseline for the benchmark that uses a specialized grid-based architecture for the task and learns a language-conditioned policy with PPO (Schulman et al., 2017). As seen in Figure 7, Dynalang achieves higher performance and learns more efficiently than EMMA, IMPALA and R2D2. While other methods fail to fit S3 at all, our method learns to interpret the manuals to achieve non-trivial performance on the most challenging stage, further supporting **H3**.

### 4.4 VISION-LANGUAGE NAVIGATION: INSTRUCTION FOLLOWING

To evaluate how Dynalang performs in more complex environments, we apply it to the popular Vision-Language Navigation (VLN) (Anderson et al., 2018) benchmark. Agents must navigate Matterport3D panoramas captured in real homes (Chang et al., 2017), following crowd-sourced natural language instructions that indicate where the agent should navigate to, such as "Go past the end of the bed to the door. Enter the hallway,..." We focus on the more challenging variant, VLN in Continuous Environments (VLN-CE) (Krantz et al., 2020), in which agents take low-level discrete actions (left, forward, ...) rather than relying on a waypoint navigation graph as in the original VLN task. In this task, our goal is to demonstrate that Dynalang can learn policies in this challenging instruction-conditioned RL setting while interpreting instructions as predicting future rewards.

Each episode randomly samples a language instruction and corresponding scene from the training dataset, which is comprised of 10,819 unique instructions total. The agent is trained with a dense reward based on relative positions to the current goal, a success reward when taking the `stop` action at the correct location, and a penalty otherwise. Compared to the model-free R2D2 baseline, Dynalang succeeds at a significantly higher portion of the training instructions, supporting **H4**. While Dynalang successfully learns to ground instructions from scratch, performance is not yet competitive with state-of-the-art VLN methods (many of which use expert demonstrations or specialized architectures), and further work is needed to close the gap.

### 4.5 LANGROOM: EMBODIED QUESTION ANSWERING

Finally, we show how Dynalang can also *generate language* in the same framework. On the other benchmarks, language is used to inform agents' future predictions about the world, but perception

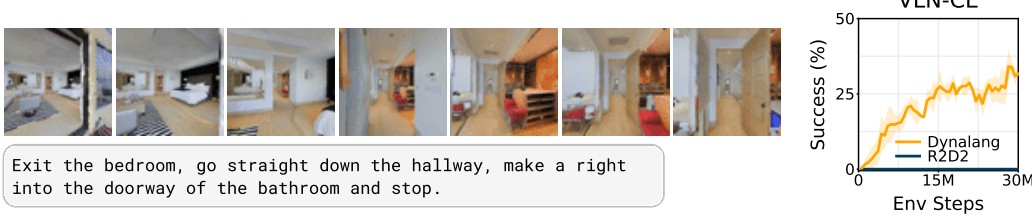

Figure 8: VLN-CE results. **(left)** A portion of a trained agent trajectory, given the instruction "Exit the bedroom, go straight down the hallway, make a right into the doorway of the bathroom and stop". **(right)** Success rate during RL training, averaged across 3 seeds for Dynalang and 2 seeds for R2D2.

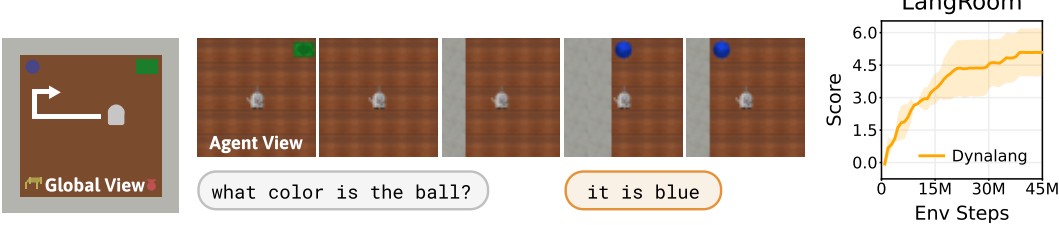

Figure 9: LangRoom results. **(left)** A real trajectory from a trained agent. The agent learns to take information-gathering actions from reward. When asked "what color is the ball?" the agent walks to the corner with the ball and generates the tokens "it is blue." **(right)** Training curve. The agent learns to answer more questions accurately.

can also inform future predictions about *what might be said*. For example, agents could predict that they will hear descriptive utterances such as "the stove is on" that are consistent with its own observations of the burner producing flames. We introduce the LangRoom embodied question answering environment to demonstrate a proof-of-concept of this capability. We expand the action space to include language by allowing the agent to output one language token per timestep as an action. The environment contains a room with objects with fixed positions but randomized colors. The language observations from the environment are *questions* "what color is the <object>?." The agent must move to the object and emit a language action saying the correct color.

As shown in Figure 9, the agent learns to answer more questions correctly with task reward, supporting **H5**. We show an example trajectory demonstrating that the agent has learned to take information gathering actions to observe the color of the object and generate text consistent with the world state.

### 4.6 Text-only Pretraining

Dynalang can be pretrained on single-modality data by zeroing out the other modality and action inputs. This provides a way for RL agents to benefit from large-scale offline data in a single architecture. To evaluate this capability, we pretrain Dynalang from scratch on (1) **in-domain text** with manuals from Messenger S2 games (2) **domain-general text** with TinyStories (Eldan & Li, 2023), a dataset of 2M short stories generated by GPT-3.5 and GPT-4. We evaluate on Messenger S2, where models that learn to embed one-hot token observations from scratch struggle to learn the complex language in S2 without pretraining on S1. We use the T5 vocabulary and compare S2 task performance with learned embeddings to using pretrained T5 embeddings, training all methods from scratch on S2 without initializing from S1. In Appendix E.1, we show how Dynalang is able to benefit from offline pretraining on text-only data. Even a small amount of in-domain text closes much of the gap between training text embeddings from scratch and using T5 embeddings. Furthermore, pretraining on TinyStories *exceeds* the final performance of using T5 embeddings, likely because pretraining allows the model to learn text dynamics offline rather than during environment interaction.

Although the model is not trained explicitly to do language modeling except through next-representation prediction, we can generate language from the world model by doing rollouts in latent space and reconstructing the token from the latent representation. In Appendix E.2 we show the model's preliminary language generation capabilities after pretraining on TinyStories, which suggest that Dynalang could potentially be trained and used as a language model.

## 5 Discussion

**Limitations** Our recurrent architecture may make optimization challenging in extremely long horizon environments. Our design decision to interleave vision and language tokens one-to-one allows the agent to act while communicating but may cause sequence length to be the bottleneck for learning in some tasks. While Dynalang can generate text, the generation quality is not competitive with pure language models and further work will be needed to close that gap.

**Conclusion** We present Dynalang, an agent that grounds language to visual experiences, actions, and rewards through future prediction as a rich self-supervised objective. Dynalang learns to act based on various types of language across a diverse range of tasks, often outperforming model-free methods that struggle with increased language complexity. The ability to pretrain on video and text without actions or rewards suggests that Dynalang could be scaled to large web datasets, paving the way towards a self-improving multimodal agent that interacts with humans in the world.

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

## A  WORLD MODEL LEARNING

**Representation Learning**   The discrete codes $z_t$ are vectors of one-hot categoricals that are sampled during the forward pass and optimized using straight-through gradients on the backward pass (Bengio et al., 2013; Hafner et al., 2020).

**Two-hot Reward Prediction**   We follow DreamerV3 in predicting rewards using a softmax classifier with exponentially spaced bins that regresses the twohot encoding of the real-valued rewards and in clipping the regularizer at 1 free nat (Kingma et al., 2016). The two-hot regression decouples the gradient scale from the arbitrary scale of the rewards and free nats prevent over-regularization, known as posterior collapse.

## B  ACTOR CRITIC LEARNING

Because we optimize the policy from imagined rollouts, all involved quantities are predictions rather than environment observations. For simplicity, we omit the hats from the notation now and e.g. write $z_t$ instead of $\hat{z}_t$. To train the actor and critic networks, we predict a sequence of $T = 15$ representations $z_t$ by sampling from the world model and the actor network. The sequences start at all representations computed from the world model training step. From a sequence of representations $z_t$ and recurrent states $h_t$, we fill in the rewards $r_t$ and episode continuation flags $c_t$ by applying their two MLPs, without invoking the image or language decoders. Given the quantities, we compute a $\lambda$-return (Sutton & Barto, 2018) that estimates the discounted sum of future rewards:

$$R_t = r_t + \gamma c_t\Big((1 - \lambda)V(z_{t+1}, h_{t+1}) + \lambda R_{t+1}\Big) \qquad R_T \doteq V(z_T, h_T) \qquad (1)$$

The return estimate $R_t$ serves as a prediction target for the critic network, which uses discrete regression using a categorical cross entropy loss towards the twohot encoded targets. The actor network is trained to maximize the return estimates subject to an entropy regularizer on the action distribution:

$$
\begin{aligned}
\mathcal{L}_V &= \text{catxent}(V_t(h_t, z_t), \text{sg}(\text{twohot}(R_t))) \\
\mathcal{L}_\pi &= -\,\text{sg}(R_t - V(z_t, h_t))/\max(1, S)\log\pi(a_t \mid h_t, z_t) - \eta\,\text{H}\big[\pi(a_t \mid h_t, z_t)\big]
\end{aligned} \qquad (2)
$$

To trade off the two actor loss terms without having to tune hyperparameters, the actor loss normalized returns that exceed a magnitude of 1 are normalized by an exponential moving average of the 5th to 95th percentile range of returns, $S = \text{ema}(\text{per}(R_t, 95) - \text{per}(R_t, 5))$. When interacting with the environment, we choose actions by incorporating the new observation into the world model representation and then sampling from the actor network.

## C  DETAILED RELATED WORK

**Language and Embodied Agents**   Language can be used in embodied settings in a variety of ways (Luketina et al., 2019). In instruction following, agents must interpret language specifications of high-level goals or step-by-step guidance (Branavan et al., 2010; Andreas & Klein, 2015; Anderson et al., 2018; Shridhar et al., 2020a; Lynch & Sermanet, 2021). Language can also be used as an abstraction to assist learning or decision-making, e.g. for planning by decomposing high-level tasks into low-level subgoals (Andreas et al., 2017; Jiang et al., 2019; Ahn et al., 2022; Huang et al., 2022a; Li et al., 2022; Sharma et al., 2021). Instead of planning in language, our model treats language as another modality in observation space and plans in latent space. Finally, language can be used to describe the world, e.g. to enable semantic exploration (Mirchandani et al., 2021; Tam et al., 2022; Mu et al., 2022; Du et al., 2023b), to communicate domain knowledge (Eisenstein et al., 2009; Branavan et al., 2010; Narasimhan et al., 2018; Zhong et al., 2020; Fan et al., 2022), or as feedback from the environment (Huang et al., 2022b). Our work investigates how to unify these settings so that agents can learn from all kinds of language they might encounter in the world, including instructions and descriptions. While most of these works directly condition policies on language to generate actions (model-free), our algorithm uses language for future prediction, learning a world model that is then used for planning and acting.

**Multimodal Models**    Developing agents that can leverage both vision and text observations requires training multimodal models. Previous works develop vision-language models (VLMs) by augmenting LLMs with visual encoders  (Alayrac et al., 2022; Li et al., 2023b; Chen et al., 2022b; Guo et al., 2023) or training models jointly over all modalities (Lu et al., 2022) However, because VLMs are prohibitively expensive to query and finetune, recent work on using VLMs as policies has focused on supervised learning from demonstrations (Driess et al., 2023; Jiang et al., 2022), rather than using them in embodied agents that can learn online. Reed et al. (2022) trains a multimodal embodied agent across various tasks, modalities, and embodiments by additionally learning to generate actions. Perhaps most similar, a recent line of work (Du et al., 2023a; Yang et al., 2023) trains text-conditioned video models for planning. Their approach works by using a video model trained on expert demonstrations to generate a video plan conditioned on a text instruction, and then imputing the actions to execute that plan. From a generative modeling perspective, our approach differs in that it also learns to generate language instead of being solely input text-conditioned, enabling text-only pretraining, interactive dialogue, and future possibilities for learning shared representations of text/video. Additionally beyond both VLMs and text-conditioned video models, our approach enables learning from online experience in addition to offline pretraining, allowing agents to improve their behaviors and understanding of the world autonomously rather than being inherently limited to an offline dataset.

**Decision-making with Large Language Models**    Large language models (LLMs) learn about the world via next-token prediction on web-text, implicitly modeling world state (Li et al., 2021; 2023c) and relations between concepts (Piantadosi & Hill, 2022). When acting in purely text-based or symbolic environments, language models can be used as complete world models (Ammanabrolu & Riedl, 2018; Singh et al., 2021). In visual environments, LLMs are not grounded to real environment observations and cannot directly take actions, unless observations are translated to text (Shridhar et al., 2020b; Huang et al., 2022b; Dasgupta et al., 2023). However, representing visual inputs as text is inherently low bandwidth. Additionally, while LLMs can be used as a prior over actions or observations (Li et al., 2023a), they are difficult to update with feedback from the environment except in limited cases (Carta et al., 2023; Dagan et al., 2023). In contrast, we learn a single multimodal world model from experience with autoregressive prediction on both text and images (predicting both modalities in the future from both modalities as input), thus grounding language to *experience* (Bisk et al., 2020). Our model can also be trained on text-only data as a language model or video-only data as a video prediction model.

# D    ENVIRONMENT DETAILS

## D.1    HOMEGRID

The HomeGrid environment is a grid with different objects, receptacles, and rooms. Agents receive pixel observations of 3x3 grid cells centered on the current agent position. The action space is: movement (`left`, `right`, `up`, `down`), object interaction (`pick up`, `drop`), and trash bin interaction (`get`, `pedal`, `grasp`, `lift`). The agent can carry one object in its inventory by executing the `pick up` action in front of an object or the `get` action in front of a trash bin with an object inside. There are three rooms (living room, dining room, kitchen) indicated by different flooring textures, three possible trash bin types with different colors (blue recycling, black trash, green compost) and four possible trash object types (bottle, fruit, papers, plates). Trash bins can be open, closed, or knocked over (represented visually as toppled over sideways). Each trash bin can be opened with a specific action that is randomly selected from {`pedal`, `grasp`, `lift`} in each episode. If agents apply the wrong action on a bin, it becomes broken and cannot be interacted with further until reset by the environment. When a trash bin is open, one object can be dropped into the bin with the `drop` action and the current object in the bin (if any) can be retrieved into the agent's inventory with `get`.

For each episode, the environment is randomly initialized with two objects and two trash bins in random positions. Trash bins are initialized in the open state with probability 0.5. One bin is irreversibly broken if the wrong action is applied and the other bin is reset after 5 timesteps if broken. At each timestep, each object is moved to a new position with probability 0.05 and new objects are spawned with probability 0.1*`num_remaining_unique_objects` at a random position.

In our experiments, agents are evaluated on setups with different language inputs: task instructions, task instructions + dynamics, task instructions + future observations, and task instructions + corrections. Language for each type is generated with templates from the underlying environment state, with the following semantics:

**Tasks**

- `find the [object/bin]`: the agent will receive a reward of 1 if it is facing the correct object / bin
- `get the [object]`: the agent will receive a reward of 1 if it has the correct object in inventory
- `put the [object] in the [bin]`: the agent will receive a reward of 1 if the bin contains the object
- `move the [object] to the [room]`: the agent will receive a reward of 1 if the object is in the room
- `open the [bin]`: the agent will receive a reward of 1 if the bin is in the open state

**Future Observations**: descriptions of environment state the agent may observe in the future

- `[object/bin] is in the [room]`: the object or bin is in the indicated room
- `i moved the [object] to the [room]`: the object has been moved to the room
- `there will be [object] in the [room] later`: the object will spawn in the room in five timesteps

**Dynamics**: descriptions of environment transitions

- `[action] to open the [bin]`: the indicated action is the correct action to open the bin

**Corrections**: task-specific feedback about the agent's current trajectory

- `no, turn around`: the agent's distance to the current goal object or bin (given the task) has increased compared to the last timestep

Language is provided to the agent one token per timestep. All language are provided while the agent acts and the environment state is changing, except for dynamics descriptions (which apply to the whole episode). For dynamics descriptions, we randomly shuffle all possible descriptions and input them to the agent in sequence up to a maximum of 28 tokens while the agent is fixed in place. For language provided during the episode, on each timestep, if there is not currently an utterance being provided to the agent, either (1) the task instruction is repeated, every 20 timesteps (2) an utterance describing one of the events that occurred at this timestep is provided (i.e. objects moved or spawned) (3) a description of future observations or dynamics is provided (4) a correction is provided, with probability 0.1. If there is a new task instruction (i.e. the agent just completed the last task), any currently streaming sentence will be interrupted and the agent will immediately receive the tokens of the new instruction. All evaluation setups share the same underlying environment dynamics and parameters (e.g. each trash bin must be operated with the correct action even if the agent does not receive hints about dynamics).

## D.2 MESSENGER

Language in Messenger is generated from human-written templates, resulting in diverse sentences with multiple ways of referring to each entity and a total vocabulary size of 1,125. Observations are presented as a symbolic grid of entity IDs, and the agent takes discrete actions to move. We input the manual into the world model token-by-token before the episode begins.

The EMMA baseline provides a gridworld-specific inductive bias that each text token should map to some region in the current observation, and assumes that the model has access to the spatial locations of entities in the scene. As in the original benchmark, we initialize all models from the converged model trained on the previous game stage.

## D.3 VLN-CE

The best-performing methods on VLN-CE use expert demonstrations (An et al., 2023) or train navigation-specialized hierarchical agents (Krantz et al., 2021). The VLN-CE training set consists of

10,819 unique natural instructions total, spread across 61 scenes. The instruction and corresponding scene are randomly sampled per episode. In addition to language, the agent observes an egocentric RGB and depth image at each timestep. Agents have access to discrete low-level actions (moving forward 0.25 meters, turning left or right 15 degrees), as well as a `stop` action. Crucially, the agent must learn to take the `stop` action when it thinks it has reached the goal to indicate that it recognizes the goal position. This makes the task more challenging, as the agent must learn to only terminate the episode at the appropriate goal locations. The agent receives a dense reward at every timestep based on the delta in position from the goal. Following (Krantz et al., 2021), we provide an additional success reward of 1000 when the agent takes the `stop` action at the correct location, and a penalty of −10 when the agent takes the `stop` action elsewhere.

## D.4 LANGROOM

In LangRoom, the environment contains four objects in the corners of a room. The positions of the objects are fixed but the colors are randomized. The action space for the agent includes the four cardinal movement actions, stay, and 15 tokens that the agent can say. The language observations from the environment are *questions* "what color is the <object>?" followed by a random silence duration (allowing the agent to find out the answer), followed by the answer "it is <color>". After each question and answer, the colors are randomized and the environment asks a new question, up to a fixed episode length of 200 timesteps. Agents are rewarded +1 for saying the correct "<color>" token at the same timestep that the environment produces the "<color>" token, −0.1 for saying the wrong color at that timestep, −0.01 for speaking at other timesteps, and 0 for saying nothing. The agent only has a partial view over the environment, so it must move to the object before the environment starts prompting it for the answer.

# E    TEXT PRETRAINING

## E.1    PRETRAINING ON TEXT-ONLY DATA IMPROVES DOWNSTREAM PERFORMANCE

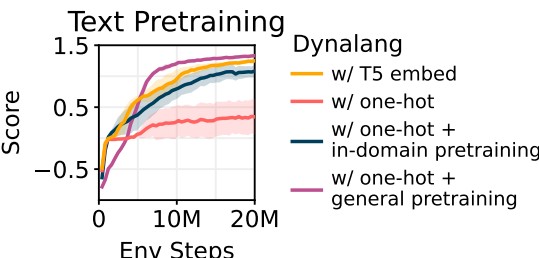

Figure E.1: One-hot token encodings underperform pretrained embeddings on S2, but pretraining Dynalang with a small amount of text-only data closes much of the gap.

## E.2    TEXT GENERATION SAMPLES

Dynalang is not explicitly trained on the language modeling objective, but we can still generate text from the model by sampling rollouts from the world model and decoding the token from the latent representation at each timestep. Here, we show sampled 10-token generations conditioned on a prefix of 50 tokens for validation examples in TinyStories.

---

**Prompt**: Once upon a time, in a big forest, there lived a rhinoceros named Roxy. Roxy loved to climb. She climbed trees, rocks, and hills. One day, Roxy found an icy hill.
**True**: She had never seen anything like it before. It
**Samples**:
She wanted to climb down the hill. friends and
It was a steep tree, but she was not
She wanted to own it, but it was too hot
She thought it would be fun banana scary, andffy
She wanted to skip through the. But Once upon

---

**Prompt**: Once upon a time, there was a thoughtful girl named Sue. Sue loved to help her mom around the house. One day, her mom asked her to wipe the table after they ate their lunch. Sue was happy to help. As
**True**: Sue was wiping the table, she saw
**Samples**:
they her big room. playly remembered her
she was placing,,, she saw a
she got being, she saw hera all she on
she was organizing, she saw the pin case in the
she was their best delicate turkey on, she saw

---

**Prompt**: Once upon a time, there was a little girl named Lucy. She had a pet cat named Tom. They loved to play together in the big green park near their house. One sunny day, they went to the park to play.
**True**: While playing, Tom saw a big s
**Samples**:
 Once upon a time, there was scarf
 " Jenny, you are my sweet. You must
 Once heard a kind girl and asked Mom to
 When taking a small, thin thing he
 The. lesson its if can improve and

---

**Prompt**: Once upon a time, there was a little boy named Tom. He loved to play with his red ball. One sunny day, Tom went outside to play with his ball in the land near his home. Tom kicked the ball high in
**True**: the sky. The ball went far, far away
**Samples**:

---

the sky and ity it." Once day,
the air and loved then it rain outside. We can
the sky, but was enormous diary to with baby
the sky. red ball went and all game,
the air and ran after to catchMoo. His was

---

**Prompt**: Once upon a time, there was a girl named Mia. Mia loved her jewelry. She had a big box full of pretty things. She liked to wear them all day. But at night, she had to sleep. One
**True**: day, Mia met a talking cat named
**Samples**:
day, shea was mad. She did not want
night, shea socks out wanted to hurt up.
day, shea could not find her skirt dress She
day, hera's mom came to her.
day, Miaa fell her hair could. It

---

**Prompt**: Once upon a time, there was a little boy named Tom. Tom had a special belt that he loved to wear. One day, he could not find his belt and felt very sad. Tom's mom saw him and
**True**: asked, "Why are you sad, Tom?"
**Samples**:
frustrated and asked him what was rude. Once upon
asked, "Why are you sad, Tom?"
asked, "Howeny, I did, get
said, "Don't worry, Tom. We
said, "To tree, you look be in

---

**Prompt**: Once upon a time, in a small house, there lived a kind and honest girl named Lily. She loved to bake cakes for her friends and family. One day, she made a big, yummy cake for her best friend
**True**: 's birthday. Lily carefully put the cake
**Samples**:
, Ben. Tom went Mike opened the and,
, Tom. Oneo decided the biggest ow
, Tim. Once upon a time, there
, Tim. lady.  and Lily
, Tom. Once upon a time, there

---

**Prompt**: One day, a young boy named Tim found a dull, round rock. He picked it up and looked at it. He thought it was not very fun, but he took it with him to the park. At the park, Tim
**True**: saw a girl named Sue. She had
**Samples**:
he met. favorite friend He put it his to
met a girl named Sue. Sue saw the ball
saw a stick top Sam. He kept playing with
played with his friends and but they friends!" Li
met a girl named Lily. ly saw

---

**Prompt**: Once upon a time, there was a little boy named Tim. Tim loved candy more than anything else. One day, Tim saw a big candy store. He was very happy and ran to the store. Inside the store, Tim met
**True**: a strange man. The man said, "
**Samples**:
a nice lady named Sue. The thing the
a tall named Max. that the clever
a girl friend named Sue. They said, "
a big dog named Theffy. said
a new prize car two cars. things.

---

**Prompt**: Once upon a time, there was a big, heavy alligator. He lived near a small pond. He was very hungry and wanted to eat something. One day, a little bunny came close to the
**True**: pond. The alligator saw the bun
**Samples**:
flower. The bunny said, "Hello
kitchen. He thisly and said, "This
bunny and askede "Do, smell you
sunflower. The bun said, "Stop, sunset
bunny. The bunny said, "

---

## F  ADDITIONAL ANALYSIS

### F.1  QUALITATIVE ANALYSIS

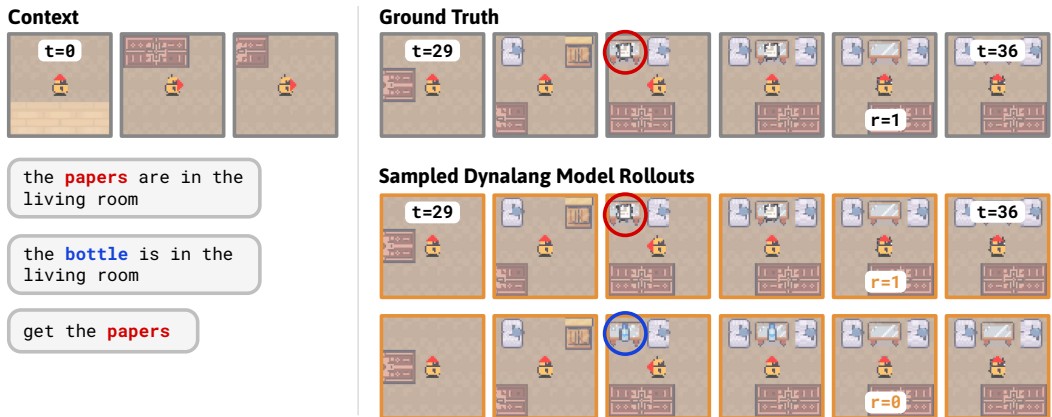

Figure F.1: Imagined rollouts from the world model. Conditioned on a language description, the task, and the same action sequence, we sample rollouts of the world model's imagined trajectories. Since the papers and bottle can be in any of multiple possible locations in the living room, the model samples exhibit uncertainty over the possible futures. In one rollout (top), the agent predicts the papers are on the table and correctly predicts it will get rewarded for picking it up. In the second rollout (bottom), it predicts that the bottle is on the table and that it will not get rewarded.

Figure F.1 shows that we can interpret what the model has learned by rolling out the world model state into the future and reconstructing observations from the latent state, conditioned on some history. We can see that the model represents the information and correctly grounds it to observations: given the information that the papers and bottle are in the living room, different samples from the world model represent different possible futures, both of which are consistent with the text. The model also correctly predicts that in the future where the papers are on the table, it will receive a reward of +1 for doing a pickup action, and that it will not be rewarded if it picks up the bottle.

### F.2  ABLATION: TOKEN EMBEDDINGS VS. SENTENCE EMBEDDINGS

**Token representations outperform sentence representations.**  Figure F.2 shows that consuming one sentence of the manual per timestep causes the agent to learn much more slowly, compared to our model which reads one token per timestep. We use embeddings from the Sentence Transformers `all-distilroberta-v1` model (Reimers & Gurevych, 2019). We hypothesize that the sentence encoder output could be a lossy bottleneck, making it difficult for Dynalang to extract information from the text particularly when the sentences contain a lot of information.

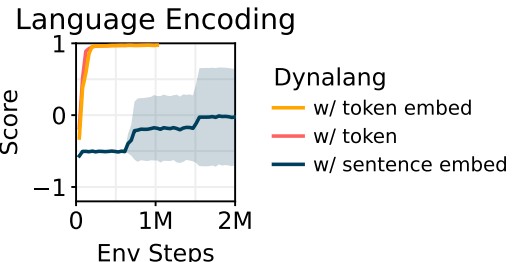

Figure F.2: Sentence embeddings lead to much slower learning, even on S1 where both one-hot and pretrained token encodings quickly reach ceiling performance.

## G    HOMEGRID TRAINING CURVES

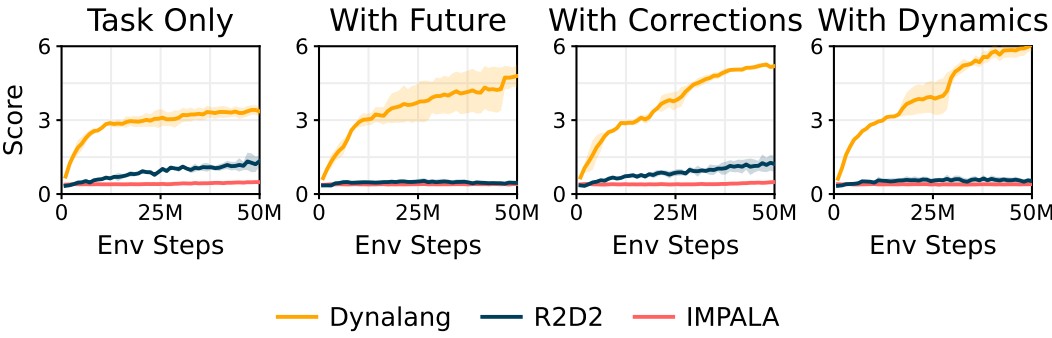

Figure G.1: HomeGrid training curves.

## H    ADDITIONAL BASELINE EXPERIMENTS

### H.1    TOKEN VS. SENTENCE EMBEDDINGS FOR BASELINES

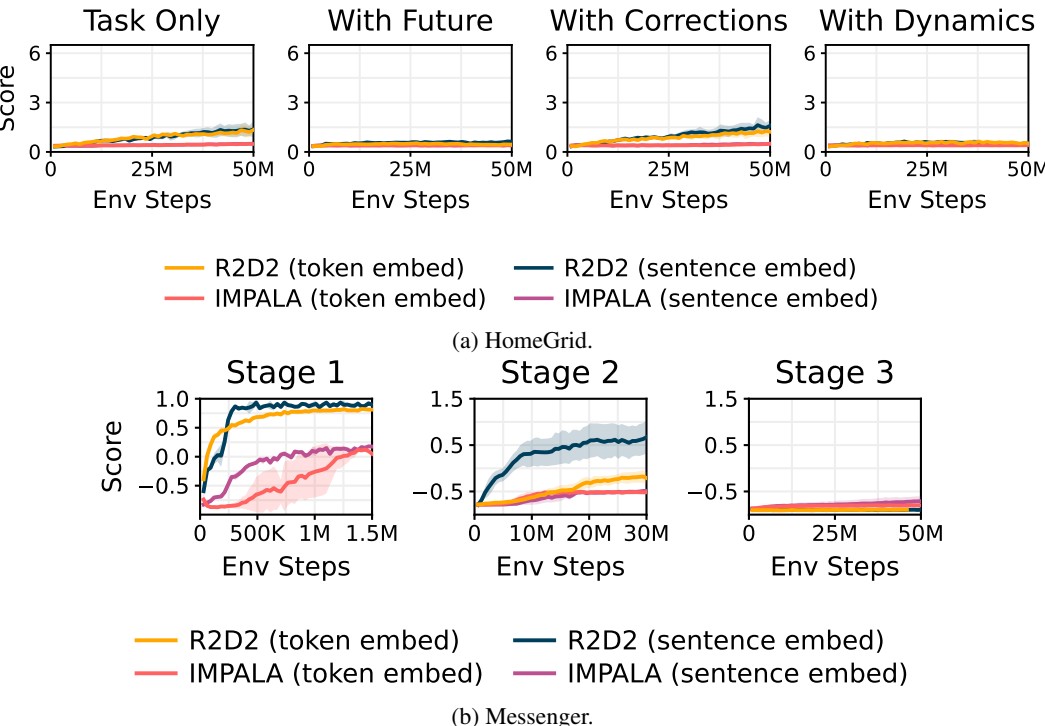

Figure H.1: Token vs. sentence embedding performance for IMPALA and R2D2 on all tasks, averaged across 3 seeds. Sentence embeddings help R2D2 perform better on Messenger S1 and S2 but does not help consistently across tasks and methods.

## H.2 MODEL SCALING FOR BASELINES

We find that scaling the baseline R2D2 and IMPALA models does not improve their performance. Stage 2 runs were initialized from scratch.

| Model Size | LSTM hidden size | Language MLP size | CNN hidden size | Policy/Value Hidden Size |
|---|---|---|---|---|
| **1.7M** | 256 | 256 | [16, 32, 32] | None (linear) |
| **10M** | 1024 | 512 | [16, 32, 32] | [512] |
| **37M** | 2048 | 1024 | [64, 64, 64] | [1024, 1024] |

Table H.1: R2D2 architecture sizes for model scaling experiment.

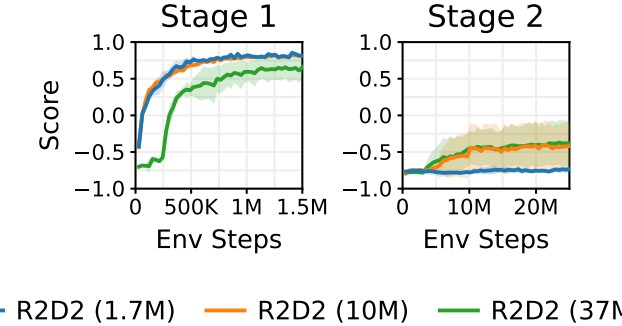

Figure H.2: Model scaling curves for R2D2.

| Model Size | LSTM hidden size | Language MLP hidden size | CNN hidden size | Policy/Value Head |
|---|---|---|---|---|
| **1.5M** | 512 | [64] | [16, 32, 32] | None (linear) |
| **8.8M** | 1024 | [512] | [16, 32, 32] | [512] |
| **34M** | 2048 | [1024] | [16, 32, 32] | [1024, 1024] |

Table H.2: IMPALA architecture sizes for model scaling experiment.

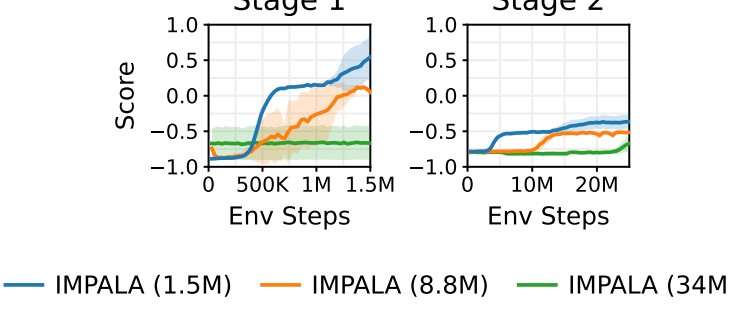

Figure H.3: Model scaling curves for IMPALA.

### H.3    AUXILIARY RECONSTRUCTION LOSS FOR BASELINES

We tried adding an auxiliary loss for reconstructing the visual and language observations at the current timestep. The loss was implemented by adding a linear layer that predicts each auxiliary target from the LSTM hidden state. The loss used is MSE (for continuous values) or cross-entropy (for discrete language vocab tokens). The auxiliary loss was added to the RL loss with a loss scale of 1. This did not meaningfully change performance.

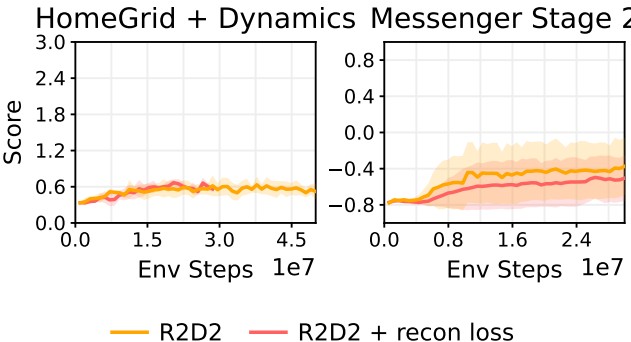

Figure H.4: Model-free R2D2 performance with an auxiliary reconstruction loss.

# I  MODEL AND TRAINING DETAILS

## I.1  BASELINE HYPERPARAMETERS

| | HomeGrid | Msgr S1 | Msgr S2 | Msgr S3 | VLN |
|---|---|---|---|---|---|
| Total model parameters | 27M | 10M | 10M | 10M | 10M |
| Language inputs | One-hot | T5 Embed | T5 Embed | T5 Embed | T5 Embed |
| Vocabulary size | 32100 | n/a | n/a | n/a | n/a |
| Language MLP layers | 1 | 1 | 1 | 1 | 1 |
| Language MLP units | 512 | 512 | 512 | 512 | 512 |
| Image input | Pixel | Symbol | Symbol | Symbol | Pixel |
| Image size | (64, 64, 3) | (16, 16, 17) | (16, 16, 17) | (16, 16, 17) | (64, 64, 3) |
| Replay ratio | 7 | 7 | 7 | 7 | 7 |
| Batch size | 32 | 64 | 16 | 16 | 8 |
| Unroll length | 100 | 100 | 100 | 100 | 100 |
| LSTM recurrent units | 1024 | 1024 | 1024 | 1024 | 1024 |
| Learning rate | 4.8e-4 | 4.8e-4 | 4.8e-4 | 4.8e-4 | 4.8e-4 |
| Buffer Size | 1000 | 1000 | 1000 | 1000 | 1000 |
| Env steps | 50M | 1M | 25M | 50M | 30M |
| Number of envs | 80 | 80 | 80 | 80 | 5 |

Table I.1: Model hyperparameters and training information for the R2D2 baseline.

| | HomeGrid | Msgr S1 | Msgr S2 | Msgr S3 |
|---|---|---|---|---|
| Total model parameters | 10M | 9M | 9M | 9M |
| Language inputs | One-hot | T5 Embed | T5 Embed | T5 Embed |
| Vocabulary size | 32100 | n/a | n/a | n/a |
| Language MLP layers | 1 | 1 | 1 | 1 |
| Language MLP units | 512 | 512 | 512 | 512 |
| Image input | Pixel | Symbol | Symbol | Symbol |
| Image size | (64, 64, 3) | (16, 16, 17) | (16, 16, 17) | (16, 16, 17) |
| Batch size | 16 | 64 | 64 | 64 |
| LSTM recurrent units | 1024 | 1024 | 1024 | 1024 |
| Learning rate | 3e-4 | 3e-4 | 3e-4 | 3e-4 |
| Env steps | 50M | 1M | 25M | 50M |
| Number of envs | 80 | 80 | 80 | 80 |

Table I.2: Model hyperparameters and training information for the IMPALA baseline.

## I.2 DYNALANG HYPERPARAMETERS

We use the default model hyperparameters for the XL DreamerV3 model unless otherwise specified below. For VLN, we use a larger GRU deterministic state and a bottleneck layer of size 1024 between timesteps. To process both one-hot and embedding language inputs, we use a 5-layer MLP with 1024 MLP units in each layer. All models were trained on NVIDIA A100 GPUs.

|  | **HomeGrid** | **Msgr S1** | **Msgr S2** | **Msgr S3** | **VLN** | **LangRoom** |
|---|---|---|---|---|---|---|
| Total model parameters | 281M | 148M | 148M | 148M | 268M | 243M |
| Language inputs | One-hot | T5 Embed | T5 Embed | T5 Embed | T5 Embed | One-hot |
| Vocabulary size | 32100 | n/a | n/a | n/a | n/a | 15 |
| Language MLP layers | 5 | 5 | 5 | 5 | 5 | 5 |
| Language MLP units | 1024 | 1024 | 1024 | 1024 | 1024 | 1024 |
| Image input | Pixel | Symbol | Symbol | Symbol | Pixel | Pixel |
| Image size | (64, 64, 3) | (16, 16, 17) | (16, 16, 17) | (16, 16, 17) | (64, 64, 3) | (64, 64, 3) |
| Train ratio | 32 | 64 | 64 | 32 | 32 | 16 |
| Batch size | 16 | 16 | 24 | 24 | 8 | 16 |
| Batch length | 256 | 256 | 512 | 512 | 256 | 64 |
| GRU recurrent units | 4096 | 4096 | 4096 | 4096 | 8192 | 6144 |
| Bottleneck units | n/a | n/a | n/a | n/a | 1024 | 2048 |
| Env steps | 50M | 1M | 25M | 50M | 30M | 45M |
| Number of envs | 66 | 16 | 16 | 66 | 8 | 4 |
| Training time (GPU days) | 3.75 | 2.5 | 16 | 24 | 16 | 2 |

Table I.3: Dynalang hyperparameters and training information for each environment.

