# OpenReview forum: "Learning to Model the World with Language"
_ICLR.cc/2024/Conference — Submitted to ICLR 2024_

### Official Review · Reviewer_PbML · 2023-10-25

**Soundness:** 3 good
**Presentation:** 2 fair
**Contribution:** 2 fair
**Rating:** 5
**Confidence:** 3

**Summary:**

The authors argue that an RL agent should use language to predict the next state of the world, which will empower them with the ability to understand the world and thus generate a better policy, instead of directly learn to map language into actions. They propose to build a world model that can predict future language, video and rewards, and demonstrate that training an agent with the world model achieves better performance over other baselines.

**Strengths:**

1. The motivation is interesting and convincing. The large language models learn rich knowledge about the world by only predicting the next word, so it is reasonable to hypothesize that utilizing language for future prediction is a better way to help agent understand the world.
2. Experimental results show that the proposed method outperforms the baselines.

**Weaknesses:**

Although the motivation is promising, the method and experiments do not support the claim.
1. It is confusing that the authors use a multimodal model including both text and images to demonstrate the idea of using language to model the world. Images also convey general knowledge and describe the state of the world, then why can't we also model the world with images / videos? The authors should provide more evidence to demonstrate the unique importance of language to support their claim.
2. The method proposed in this paper is quite like the Dreamer V3 model [1] with additional text input. In Dreamer V3 paper, they have already demonstrated the effectiveness of their method, and the authors seem to simply apply it on environments that include text. Then, how to clarify that the improvements come from the the model architecture itself or the text part? There are no experiments to demonstrate this. Notice that the author even don't compare with other model-based methods that are more similar to their proposed method, although they claim they compared with them in the introduction.

[1] Hafner et al. Mastering Diverse Domains through World Models. arXiv 2023.

**Questions:**

The paper mentioned that at one time step only one text token will be included in the observations and the model output. I don't quite understand the setting here. If this is the case, then the setting is quite limited and it also conflicts with the example "I put the bowl away" you use in the introduction?

---

> ### Author Response · Authors · 2023-11-15
>
> Thank you for your time! It seems like there were some misunderstandings about the paper, which we hope to clarify and use as feedback to improve our paper presentation.
>
> > Comparison to DreamerV3
>
> We actually devote a whole section of the experiments to address the question of how we improve over DreamerV3 (Hypothesis H1). We apologize that this wasn't clear in the initial draft, and have updated the PDF with a much clearer presentation. **Could you please take a look and let us know if this addresses your concern that we did not compare to other model-based methods?**
>
> All of the baselines in Section 4.1 and Figure 5 are model-based baselines, where we test various ways to add language to the base DreamerV3 agent, using approaches from the multimodal literature as well as other intuitive approaches. However, these methods work significantly worse. We hope that our exploration of the design space makes it clear that our design decisions in Dynalang were non-trivial insights on top of Dreamer (given that none of these alternatives work nearly as well), and provide insight to the community on how to add language to model-based agents.
>
> > It is confusing that the authors use a multimodal model including both text and images to demonstrate the idea of using language to model the world. Images also convey general knowledge and describe the state of the world, then why can't we also model the world with images / videos? The authors should provide more evidence to demonstrate the unique importance of language to support their claim.
>
> Since we focus on agents that will act in the world, they have to use image inputs.
>
> Our paper shows that language is important over images/videos in several ways:
> - In HomeGrid, language provides additional information that is helpful to doing the task. In Figure 4 we compare using task only language with additional informative language, and the latter is better. Thus, language provides valuable information.  A purely image-based world model would not be able to use these additional types of language (and would not even be able to use language for task specification).
> - In VLN and Messenger, you can't do the task without language since language tells agents what the task is. Agents that just model visual observations would not be able to read the manual in Messenger to solve the environment or know what instruction to follow in VLN.
>
> **The idea of our paper is to demonstrate a way to build agents that is compatible with using *both* images and language to understand the world.** Language models just build a "model of the world" in text and model-based RL agents just model images, but neither of these methods on their own would be able to combine knowledge in language and visual observations like Dynalang does. Both of these modalities should be important and useful for intelligent agents in the long term.
>
> > The paper mentioned that at one time step only one text token will be included in the observations and the model output. I don't quite understand the setting here. If this is the case, then the setting is quite limited and it also conflicts with the example "I put the bowl away" you use in the introduction?
>
> We think there may be a misunderstanding about the model — the agent receives one token at each timestep, but interacts with the environment for multiple timesteps, so the agent will observe full sentences like "I put the bowl away" streaming in over multiple timesteps (similar to a language model, which also gets one token per "timestep").
>
> We compare directly to encoding one sentence per timestep with a pretrained sentence embedding model in Figure 5 and Appendix F.2, and find that our design choice vastly outperforms that method.
>
> -----
>
> **Please let us know this addresses your concerns, and if other additional information would be helpful or strengthen our work.** Thank you again for engaging with our paper!

---

> > ### Comment · Reviewer_PbML · 2023-11-21
> >
> > Thanks for the detailed respones. I raised my score to 5.
> >
> > I still feel the title is confusing / over-claiming at this point. It gives a sense that you are using pure text to model the world, while the model in the paper is using both text & image. I think your claim is sthm like using language can model the world better, so the paper may require further revision and reorganize to convey this idea more clearly.

---

> > > ### Author Response · Authors · 2023-11-21
> > > **Updated title**
> > >
> > > Thank you for raising that concern — we see now that the title leads to confusing interpretations and came up with this instead:
> > >
> > > **Understanding Language in the World by Predicting the Future**
> > >
> > > We think this definitely better represents the contribution and focus of the paper (how to learn from language in an embodied environment, and the method we propose to do so). Would changing the title to something like this address your concern?

---

> > > > ### Author Response · Authors · 2023-11-22
> > > >
> > > > Hi, as it's the last day of the discussion period, we wanted to check whether we addressed your concerns that caused you to be hesitant to recommend acceptance. We appreciate that you also thought the motivation was exciting — you were initially concerned that our experiments and setting did not support our motivation, but in light of:
> > > >
> > > > 1. Section 4.1 with comparisons to six model-based baselines (combined with our experiments on four environments outperforming model-free baselines)
> > > > 2. A title change to **Understanding Language in the World by Predicting the Future** to make clear up front what role language plays in our work/method
> > > > 3. Clarifications to your questions on the model inputs and how our experiments demonstrate that language plays a significant role over world modeling with videos only
> > > >
> > > > Would you be willing to reconsider the work holistically, or otherwise let us know how we can improve our work to better support our motivation and claims?

---

### Official Review · Reviewer_PggN · 2023-10-26

**Soundness:** 3 good
**Presentation:** 4 excellent
**Contribution:** 3 good
**Rating:** 6
**Confidence:** 4

**Summary:**

This work proposes a conditional generative model that aligns both image frames and textual instruction tokens (one at a time) to produce multimodal future representations that can encompass visual frames, textual tokens, as well as motor actions, for controlling an agent in an environment.
The proposed method is claimed to align the visual-linguistic representations better, while encouraging the models to understand the world-dynamics in a generative modeling manner.
The method is tested on four simulated embodied environments where the agents follow certain language instructions, where performance gains are reported against two off-policy RL baselines.

**Strengths:**

- The observed multimodal alignment mechanism is interesting and with experimental justification.
- The overall proposed method is neat, where the generative mechanism is a sound and interesting idea to model the visual-linguistic dynamics of the work.
- Consuming all modalities in one model as conditional generative models is neat.
- The paper is well written and easy to follow.

**Weaknesses:**

- The title is a bit over-claimed, in the sense that the proposed model is still learning to model “one environment” at a time, particularly for the action dynamics as multimodal representation generation. At least an experiment or novel method is required to learn to model some worlds (environments) and generalize to a held-out test world – this would justify the “modeling the world” parts of the claims.
- While claimed to be flexible, in many applications, the instructions of a task will only take place at the beginning of the episode while the rest is the robots’ job to accomplish the instructed tasks, where the proposed multimodal alignment will only be performed from the beginning few frames of the episode. How does the proposed method work under such conditions? E.g., how would the method benefit from such an alignment in environments such as ALFRED [1] or TEACh [2]?
- In Section 4.4, the performance of the actual SOTA models need to be reported as well, even if the proposed method is inferior to them. There are reasons why modularization and use of certain foundation models is beneficial in these long horizon complex (at least closer to) real world tasks.
- The environments, if at all except for navigation, are all quite toy-ish, where the visual observations are of fairly low fidelity. Since the proposed method heavily relies on the future representation predictions, examining the method on more realistic embodied environments would strengthen the work more.

[1] Shridhar, Mohit, et al. "Alfred: A benchmark for interpreting grounded instructions for everyday tasks." Proceedings of the IEEE/CVF conference on computer vision and pattern recognition. 2020.

[2] Padmakumar, Aishwarya, et al. "Teach: Task-driven embodied agents that chat." Proceedings of the AAAI Conference on Artificial Intelligence. Vol. 36. No. 2. 2022.

**Questions:**

- The proposed method shares some similarities with generative video-guided planning (at least at their high-levels), such as [3]. Could you elaborate more on why this is not an incremental concept on top of these works? (Also these works use supposedly much stronger generative models that can tackle more real-world visual observations.)
- What if the language instruction has a much shorter token span and the visual frames are much longer? How do they pad to each other or what would be the token used when language is exhausted out?
- Typos in “Future Prediction” of Section 3.1 – “whih” should be “which”.

[3] Dai, Yilun, et al. "Learning universal policies via text-guided video generation." NeurIPS 2023

**Details Of Ethics Concerns:**

None.

---

> ### Author Response · Authors · 2023-11-16
>
> Thank you for your constructive feedback and effort in reviewing our work! We share your excitement that our approach of modeling language and vision in a single generative model may be an interesting direction for RL.
>
> ## **Language-image alignment and flexibility**
> > While claimed to be flexible, in many applications, the instructions of a task will only take place at the beginning of the episode while the rest is the robots’ job to accomplish the instructed tasks where the proposed multimodal alignment will only be performed from the beginning few frames of the episode. How does the proposed method work under such conditions? E.g., how would the method benefit from such an alignment in environments such as ALFRED [1] or TEACh [2]?
>
> We wanted to re-emphasize that **the focus of our paper is exactly that we evaluate on different language types and alignment beyond instructions at the beginning of an episode!** For example, in HomeGrid, the agent gets language hints over the course of the episode that depend on the current state of the world — corrections like "no, turn around" when the agent is going in the wrong direction, or "I moved the plates to the kitchen" when the environment state changes in the middle of an episode. This is a major difference from almost all prior work, including the work you referenced on generative video-guided planning.
>
>
> Here is a video that shows how language and video are aligned over the course of an episode:  https://drive.google.com/file/d/1QSy-cBsrBFJi7lkowEGHbB7SIKaNkdCD/view
> - In HomeGrid, the agent receives language throughout the entire episode, including descriptions of the environment as it changes or time-sensitive feedback.
> - In Messenger, we input the entire manual at the beginning of the episode, but it is far from a simple instruction and more like a rulebook (some of the rules may or may not be relevant, and the agent has to reason over them and the visual observations). Example rulebook:
> ```
> you are being approached by a ferry which is the restricted message.
> the thief holds the a dangerous adversary and is getting away.
> the motionless ferry is the deadly adversary.
> bandit is the immovable object holding a document that is secret.
> the objective is crucial and is a stationary plane.
> the object which chases is a plane.
> the plane has an enemy that is dangerous.
> ```
>
> - In VLN, instructions are fed in from the beginning and optionally repeated throughout the episode. This is similar to instruction following in ALFRED. In ALFRED the complex instruction is broken down into substeps — this is just as easy to feed into Dynalang throughout the episode.
> - In LangRoom, the object colors are continually changing throughout the episode, so the questions and answers are time-sensitive and aligned to visual observations. This is similar to TEACh. We acknowledge their QA environment is more complex than LangRoom, but the alignment is the same: Dynalang can receive questions over time and generate answers as it explores the environment.
>
>
> > What if the language instruction has a much shorter token span and the visual frames are much longer? How do they pad to each other or what would be the token used when language is exhausted out?
>
> We hope the video link above also helps clarify here. Language is input throughout an episode rather than all at the beginning. We input a <pad> token for timesteps where there is no language.
>
> ## **Environment complexity**
> > The environments, if at all except for navigation, are all quite toy-ish, where the visual observations are of fairly low fidelity. Since the proposed method heavily relies on the future representation predictions, examining the method on more realistic embodied environments would strengthen the work more.
>
> We are also excited to test this method on more complex environments! We wanted to emphasize that our focus in this work was on **scaling up language complexity rather than visual complexity** (which is well-studied in RL) in a way that is still compatible with RL and online improvement. In this regard, our environments are much more complex linguistically than the kinds of language typically studied in RL (e.g. Messenger is symbolic but very difficult for language understanding and grounding, requiring multi-hop reasoning over long manuals and what they mean in the environment).

---

> ### Author Response · Authors · 2023-11-16
> **Relation to text-conditioned video-guided planning**
>
> We wanted to fully address your question about the relation to text-conditioned video planning methods like UniPi. This is a great point—we look forward to updating our related work with this discussion! Our approach is indeed similar in spirit but there are major conceptual and practical differences. At a high level, we focus on enabling more complex language capabilities, and this framework allows improvement from online experience instead of relying on offline expert videos.
>
> More specifically:
> - **Flexible/diverse language inputs**: Our joint language-vision sequence model enables flexible language inputs throughout an episode (e.g. instructions, manuals, interactive feedback midway through an episode), while approaches like UniPi focus on video modeling from short text prompts provided at the beginning of an episode. It's not clear that UniPi would be successful at generating video plans based on e.g. understanding a text manual. We specifically evaluate our model in settings like this with complex types of language.
> - **Language generation**: Dynalang learns to predict future language representations, whereas UniPi is only input text-conditioned. This means Dynalang can naturally be pretrained on text-only data like language models (Section 4.6 and Appendix E), opening up future possibilities for learning shared representations of text/video. This also enables interactive agents that could respond to humans in language, like in LangRoom (Section 4.5) or TEACh.
> - **Self-improvement**: Approaches like UniPi use the video prediction model to optimistically imagine a successful state sequence before filling in actions that execute that plan, which means it cannot easily be finetuned on online experience. Their task performance is thus inherently limited to the quality of expert behaviors in the pretraining video dataset. In contrast, Dynalang predicts what will happen conditioned on motor actions, which allows it to iteratively improve dynamics predictions (to better represent how the world realistically works) and the policy (to better maximize rewards).
>
> There are other practical differences like the planning method (UniPi relies on collocation-based planning which is known to fail due to over-optimism bias in stochastic or partially-observed environments [1]) and predicting in latent space (UniPi predicts future image frames, which may be overspecified), which we are happy to discuss further if you are interested!
>
> ------------------
>
> [1] Rybkin et al. 2021. Model-based Reinforcement Learning via Latent-Space Collocation.

---

> ### Comment · Reviewer_PggN · 2023-11-21
>
> Thanks for the detailed responses, where I particularly like the detailed comparisons with text-video-guided works, please expand the discussion in the paper or future submissions as well.
>
> However, my major concerns still remain.
>
> (1) Firstly, the TEACh dataset also comprises intra-episode conversations which provide just-in-time guidance. Secondly, I still do not see why this is beneficial, when in reality your language token span is usually much more coarse than visual streams (especially when the FPS is high).
>
> (2) While I like the general idea of this work, the language complexity seems a bit over-claimed here. The level of complexity should perhaps at least match that of datasets such as TEACh or RxR [1]. And that the language part of the model in this work, does not even need to exploit large-scale pretrained LMs, also justifies the language being modeled here, is not as complex as it is claimed.
>
> (3) I guess the above (2, 3) both originated and instantiated from the over-claimed title and the grand promise of **modeling the world**, while in reality the actual execution of this work is merely **modeling some synthetic action dynamics** paired with language. And furthermore, the modeling here also only supports one environment at a time. If the claim ought to be this grand, one needs to really try something that is realistic and generalizable, to justify the "modeling the world" part of the promise.
>
> In summary, I generally like and appreciate the work, should the promise be more grounded (revealing what it actually does) and/or the evaluated environments be more realistic, I would not hesitate to accept this work.
>
> [1] Ku, Alexander, et al. "Room-across-room: Multilingual vision-and-language navigation with dense spatiotemporal grounding." EMNLP 2020.

---

> ### Author Response · Authors · 2023-11-21
>
> Thank you very much for your response!
>
> 1) It's important to note that TEACh and RxR are centered around large datasets of expensive human-annotated dialogues in the environment, and agents are typically based on supervised learning on these datasets to handle more complex language.
>
> However, eventually we'd like agents to be capable of **improving both their actions and language understanding abilities online**, like people do (e.g. while exploring the world autonomously or interacting with humans). This is the question we address in our work: how can we enable learning in the world (without being inherently limited to language and behaviors in a static dataset)? This is an open question that TEACh/RxR agents *do not* address, and it is also unclear how to finetune pretrained LMs to better ground language in a visual environment. The goal of our work is to sketch out a new algorithm/method to do so, grounding language via future prediction, which is still being compatible with LM pretraining, dialogue, etc. such that future work can fully bridge the gap between these communities that work on supervised learning/rich natural language <> RL/simple language.
>
> 2) We don't mean to overclaim by using the term "modeling the world" — "world model" is standard in the RL literature and is used by others to mean "task-specific models" as well: e.g. in one of the original usages [6] they build a "world model" of a 2D car racing game. The term generally refers to model-based RL, and in RL, agents are typically single-environment (training agents that benefit from general offline data or multiple environments is an active research area).
>
>
> ## Proposed revisions
> We really appreciate the points you raised — we hope to make our claims more precise based on this discussion, updating the paper with the following:
> 1) We improve language complexity **for _agents that are capable of learning language/behavior online_** (e.g. [1-5]) but still have to bridge the gap to truly complex natural language settings like TEACh, RxR, or real human interaction. We are also excited to discuss what the remaining gaps are (e.g. intra-episode conversations, more dynamic turn-taking and dialogue, more general-domain text).
> 2) By "world model," we mean in the sense it is used in the RL community, a learned task-specific simulator of the environment. We do not address learning domain-general models of the world, but this is an important direction for future work.
>
> How do these revisions sound to you?
>
> ---------
>
> [1] DeepMind Interactive Agents Team, 2021. Creating Multimodal Interactive Agents with Imitation and Self-Supervised Learning
>
> [2] Lynch et al. 2020. Language-conditioned imitation learning over unstructured data.
>
> [3] Jiang et al. 2022. VIMA: General Robot Manipulation with Multimodal Prompts
>
> [4] Shridhar et al. 2021. CLIPort: What and Where Pathways for Robotic Manipulation
>
> [5] Du et al. 2023. Learning universal policies via text-guided video generation.
>
> [6] https://worldmodels.github.io/

---

> > ### Comment · Reviewer_PggN · 2023-11-21
> >
> > Thanks for the detailed response.
> >
> > Yes I think that is my point, the "world model" term, which is frequently used in RL domain (especially model-based RL), has a subtle difference in meanings with "modeling the world".
> > The former treats the environment as the "world" and learn/utilize the underlying dynamics, while the latter, used in this work, seems too grand at a glance (which is the precise reason that leads to my requests of cross-domain modeling/learning).
> > I see growing trend these days, especially in the LLM era, that people tend to use more flashy terms to catch the eyes, but often over-promise what the work is about.
> >
> > I'm willing to re-evaluate the whole paper when the revision is added (I didn't seem to find them in the current manuscript?).
> > Also, please have a dedicated paragraph discussing this work against the above discussed (more realistic) embodied works, and how this work can be envisioned to help them (as a future direction).

---

> > > ### Author Response · Authors · 2023-11-21
> > > **Updated title**
> > >
> > > We're actively working on the revisions and will update it shortly!
> > >
> > > Thank you for clarifying your concern about "modeling the world." We do hope to accurately convey the contribution of our paper and came up with this revised title:
> > >
> > > **Understanding Language in the World by Predicting the Future**
> > >
> > > We think this definitely better represents the contribution and focus of the paper (how to learn from language in an embodied environment, and the method we propose to do so). Would this address your concern?

---

> > > > ### Author Response · Authors · 2023-11-22
> > > > **Revision posted**
> > > >
> > > > Hi, we've posted the revised paper! Summary of changes:
> > > >
> > > > 1. We did a more thorough literature search of work along the lines of TEACh and RxR with more naturalistic language in embodied environments, and expanded Section 2 (Related Work, highlighted in green) with these references, discussing our limitations relative to these works + how we imagine our line of work fitting into these approaches in the future. We also incorporated our discussion of text-guided video planning into Appendix C (also highlighted in green).
> > > >
> > > > 2. We checked our paper to remove all instances of phrases like "modeling the world," adding a small note (highlighted in green) where we mention the phrase "world model" to clarify that we mean a simulator of the environment. We think the new title **Understanding Language in the World by Predicting the Future** is an improvement and plan to change it if you also think it is appropriate!
> > > >
> > > > Thanks for your consideration and effort in helping us improve this work!

---

> > > > > ### Comment · Reviewer_PggN · 2023-11-22
> > > > >
> > > > > Thanks for the revision, I think this is indeed better.
> > > > >
> > > > > Although I would say an even more precise title should/would be:
> > > > >
> > > > > ***"Jointly Learning Language and Environmental Dynamics via Multimodal Future Prediction"***
> > > > >
> > > > > (Here the term `language` to be more precise can be the `instruction`.)
> > > > >
> > > > > I'm raising my score to 6, per the discussion.

---

> > > > > > ### Author Response · Authors · 2023-11-22
> > > > > >
> > > > > > Thank you again for your feedback!

---

### Official Review · Reviewer_X6yV · 2023-11-01

**Soundness:** 2 fair
**Presentation:** 3 good
**Contribution:** 3 good
**Rating:** 6
**Confidence:** 4

**Summary:**

This paper addresses the challenge of enabling RL agents to comprehend and act based on complex language input. The proposed framework, Dynalang, enhances agent performance by incorporating language signals into the prediction of future states. Notably, Dynalang builds upon DreamerV3 by introducing text tokens into observations at each step. Experimental results demonstrate its effectiveness across various games, such as Homegrid, Messenger, Habbit, and LangRoom, outperforming previous language-conditioned RL baselines.

**Strengths:**

1. The paper addresses a compelling problem by enabling RL agents to understand intricate human language, expanding beyond straightforward task instructions, which is an understudied but important area in RL research.

2. The paper's writing, especially in the introduction, effectively highlights the core problem and how Dynalang provides a solution.

3. The study includes experiments across multiple game environments and consistently demonstrates improvements over existing language-conditioned RL methods.

**Weaknesses:**

1. The technical contribution is somewhat limited, primarily differing from DreamerV3 by adding text tokens to observations. A deeper exploration of Dynalang's components and their significance is needed. For example, an ablation study could help clarify the role of the language token in the world model.

2. The paper lacks a detailed ablation study that could validate the importance of each component in Dynalang. Explaining why the language token is necessary, particularly if it only serves as input for the policy network, would provide valuable insights.

3. While the paper explores various game environments, they appear simplistic. Evaluating the method on more challenging games, such as Crafter or Minecraft, would enhance the paper's credibility.

Overall, the paper presents an intriguing idea but requires further validation and clarification to strengthen its foundation. I look forward to discussing these points further in the rebuttal stage.

**Questions:**

1. How does the paper ensure that the agent can effectively follow language corrections in the Homegrid environment? Are auxiliary reward signals used to guide agent learning?

2. Could you provide more details on the training process? Is the network trained from scratch, or is the world model pre-trained?

3. Have you considered using an LLM as the core of the world model, given its strong language modeling capabilities?

---

> ### Author Response · Authors · 2023-11-15
> **Exploring design decisions in Dynalang and ablation study**
>
> Thank you for your constructive feedback and questions! We really appreciate your effort in reviewing our work.
>
> We are actively working on a more extensive response to your other questions but wanted to ask a follow-up question to the point you raised about **"exploration of Dynalang's components and their significance"** and a corresponding ablation study.
>
> We apologize that our presentation of this was not very clear in the initial draft! Section 4.1 compares ways of adding language to the DreamerV3 base agent, where we conclude that the design decisions we made in Dynalang are important and outperform a range of alternative approaches, from the previous literature as well as other intuitive approaches. We have rewritten this section to more clearly present the comparisons we made—we hope that our exploration of the design space provides insights on how to add language to model-based agents!
>
> Would you mind taking a look at the rewritten Section 4.1 and Figure 5 and letting us know to what extent they address your concern? Please let us know whether additional information would help us test our hypotheses more effectively.

---

> ### Author Response · Authors · 2023-11-19
> **Answers to additional questions**
>
> Hi, **our updated Section 4.1 potentially clarifies your main concerns about experimental validation — please let us know if any more information is needed there!**
>
> To answer the rest of your questions:
>
> **More challenging games such as Crafter or Minecraft**
>
> We are also excited to test this method on more complex environments! We wanted to emphasize that our focus in this work was on **scaling up language complexity in RL**, which is not well-studied in RL (compared to visual or task complexity). In this regard, our environments are much more complex linguistically than the kinds of language typically studied in RL (e.g. CALVIN [1] or MiniGrid [2] where the language is limited to simple instructions like "push the button"). For instance, in Messenger, agents have to solve a very difficult language understanding problem with RL rewards, requiring multi-hop reasoning over long manuals and what they mean in the environment. Future work would have to figure out how to add such language to Crafter and Minecraft, which do not natively come with language in the environment.
>
> **Language corrections in HomeGrid**
>
> > How does the paper ensure that the agent can effectively follow language corrections in the Homegrid environment? Are auxiliary reward signals used to guide agent learning?
>
> We **do not** use any auxiliary reward signals or supervision beyond task reward, which is why the result the agent can learn to ground corrections reliably (and other language, where the corresponding observation is often distant from the text) was surprising to us! Intuitively, the agent can learn to ground corrections from task reward because turning around leads to higher value states.
>
> **Pre-training**
>
> > Could you provide more details on the training process? Is the network trained from scratch, or is the world model pre-trained?
>
> The network is trained from scratch for our RL results (Sections 4.1-4.5), but we also ran experiments where we pre-trained the world model on text-only data in Section 4.6 and Appendix E and found that it improved RL performance, suggesting this would be interesting to explore in future work.
>
> **LLM as core**
>
> > Have you considered using an LLM as the core of the world model, given its strong language modeling capabilities?
>
> We'd note that the world model can also be pretrained as a language model on text-only data, so the question is perhaps more precisely whether an _existing off-the-shelf_ LLM should be used as the backbone of the model. While existing LLMs have better language understanding capabilities out-of-the-box, there are other disadvantages for RL such as being slower to train, having to modify the Transformer architecture to be stable for RL [3], and open questions such as how to adapt the model for streams of video observations. It's a very ongoing research area which we view as interesting direction to make compatible with our framework in the future!
>
> -----------
>
> [1] Mees et al. 2022. CALVIN: A Benchmark for Language-Conditioned Policy Learning for Long-Horizon Robot Manipulation Tasks
>
> [2] Chevalier-Boisvert et al. 2018. BabyAI: A platform to study the sample efficiency of grounded language learning.
>
> [3] Parisotto et al. 2020. Stabilizing transformers for reinforcement learning.

---

> ### Author Response · Authors · 2023-11-22
>
> Hi, we wanted to check whether we adequately addressed your concern about performing ablation studies to gain insight into the components of Dynalang and validating our contribution over the base Dreamer algorithm.
>
> Our hope is to propose a new way for RL agents to learn from language beyond the dominant paradigm of language-conditioned policy learning, particularly understanding more intricate human language— we hope that our empirical comparisons against both alternative model-based/Dreamer-based (Section 4.1) and model-free methods thoroughly demonstrate this point and can provide insights to others working on embodied agents + language.

---

### Official Review · Reviewer_bqw2 · 2023-11-01

**Soundness:** 2 fair
**Presentation:** 2 fair
**Contribution:** 2 fair
**Rating:** 5
**Confidence:** 2

**Summary:**

The paper proposes Dynalang, an agent that grounds language to visual experience via future prediction.

**Strengths:**

The writing of this paper is clear, and the descriptions and justifications of the methods are comprehensible.

**Weaknesses:**

This paper appears to have limited novelty, seeming more like a combination of existing techniques.

**Questions:**

What are the primary challenges addressed by the article? And what are its main contributions?

---

### Author Response · Authors · 2023-11-19
**Nearing end of discussion period**

Hello! We would like to ask the reviewers to take a look at our responses. We have updated the paper with a clearer presentation of our comparison to other model-based baselines, makeingour experimental claims more rigorous, and have further clarified our contributions in the individual responses.

Please let us know whether our response addresses your concerns or whether there is any further detail we can provide to help address them. We really appreciate your time and consideration!

---

### Meta-Review · Area_Chair_oQSA · 2023-12-06

**Metareview:**

The paper focuses on enhancing Reinforcement Learning (RL) agents' capabilities to comprehend and act based on complex language inputs. The paper integrates language signals into the prediction of future states, particularly by building upon DreamerV3 with the inclusion of text tokens.

The paper received 3 valid reviews (The review comments from Reviewer bqw2 is not well justified, and is thus discarded) , two borderline accept and one borderline reject. They acknowledge the paper's potential in advancing the field of RL with language comprehension. But the reviewers agree that further validation, clarification, and comparison with state-of-the-art models are necessary to solidify its contributions. The need for more rigorous experiments, particularly in more complex and realistic environments, and a deeper exploration of the technical aspects of Dynalang are emphasized to enhance the paper's credibility and impact.

The AC checked all the materials, and believes the weakness pointed out by the reviewers are valid. Thus, the AC decides to reject the paper.

**Justification For Why Not Higher Score:**

The reviewers found further validation, clarification, and comparison with state-of-the-art models are necessary to solidify its contributions. The need for more rigorous experiments, particularly in more complex and realistic environments, and a deeper exploration of the technical aspects of Dynalang are emphasized to enhance the paper's credibility and impact.

**Justification For Why Not Lower Score:**

N/A

---

### Decision · Program_Chairs · 2024-01-16

Reject